# Botulinum Toxin Effects on Freezing of Gait in Parkinson’s Disease: A Systematic Review

**DOI:** 10.3390/toxins16110474

**Published:** 2024-11-03

**Authors:** Nicola Tambasco, Pasquale Nigro, Alessandro Mechelli, Michele Duranti, Lucilla Parnetti

**Affiliations:** 1Neurology Department, Perugia General Hospital and University Hospital of Perugia, 06156 Perugia, Italy; lucilla.parnetti@unipg.it; 2Movement Disorders Center, Neurology Department, Perugia General Hospital and University Hospital of Perugia, 06156 Perugia, Italy; pasquale.nigro1987@gmail.com (P.N.); alessandro.mechelli@yahoo.com (A.M.); 3Department of Radiology, Perugia General Hospital, 06156 Perugia, Italy; michele.duranti@ospedale.perugia.it

**Keywords:** freezing of gait, gait, Parkinson’s disease, botulinum toxin, botulinum, motor control

## Abstract

Freezing of gait is a frequent phenomenon and can be one of the most debilitating motor impairments in Parkinson’s disease, especially in the advanced stages. It is currently defined as a brief episodic absence or any marked reduction in the forward progression of the feet, despite the intention to walk. Greater severity of freezing of gait has been associated with more frequent falls, postural instability, and executive dysfunction. However, botulinum neurotoxin is one of the most widely administered therapies for motor and non-motor symptoms, including freezing of gait, in parkinsonism. To date, the literature has had conflicting results on the use of botulinum toxin in the treatment of freezing of gait in Parkinson’s disease patients. In light of this, we reviewed the findings of past studies that specifically investigated the effects of botulinum toxin on freezing of gait in Parkinson’s disease in order to better understand this issue.

## 1. Introduction 

As a paroxysmal phenomenon, freezing of gait (FOG) is not easily assessable in outpatient settings, leading to its prevalence being underestimated. Moreover, when asked about the occurrence of FOG at home, patients and caregivers are unable to respond appropriately. According to a recent meta-analysis [1], the reported prevalence rates of FOG in Parkinson’s disease (PD) differed across studies, ranging from 5% [2] to 85.9% [3]. Moreover, a survey of 6620 PD patients observed that nearly 50% of patients experienced FOG regularly and that 80% had FOG in the advanced stages of PD [4]. Likewise, the use of specific questionnaires and targeted questions resulted in revealing FOG with a weighted average of 50.6% (with a range of 12.0–85.9%).

Regarding data on the possibility of sex as a risk factor for FOG, the evidence is contradictory: Two studies that used the Parkinson’s Progression Markers Initiative data of 393 de novo PD patients with a 4-year follow-up suggested that male sex might be an independent risk factor for FOG (HR = 1.512, *p* = 0.046) [5,6]. Other studies have supported this finding [7], including a large cross-sectional study of 6620 PD patients [4]. However, along with other authors [8,9,10,11], a recent systematic review and meta-analysis [12] reported no significant difference between the sexes for the risk of FOG development.

While age is not associated with FOG development [13,14,15], patients with longer disease duration have been reported to be at higher risk of developing FOG [15]. In fact, the overall prevalence of FOG has been reported to steadily increase with the disease progression: when assessed by specific questionnaires, the prevalence of FOG was estimated to be up to 37.9% in patients with a disease duration of 5 years or less; 48.4% between 5 and 9 years; and 64.6% in those with a disease duration of 9 years or more [1].

Research evidences that freezers have worse motor symptoms and more often show a non-tremor dominant phenotype [11,13,16]. Levodopa equivalent daily dose (LEDD) also emerged as a risk factor for developing FOG: a 100 mg increase in LEDD at baseline was reported to increase the incidence of FOG by 44.0% over a 2-year period [9,15]. Moreover, long-term pulsatile levodopa stimulation was described as a possible contributor to the development of FOG, possibly by inducing an aberrant mismatch in neuroplasticity between motor loops and cognitive and limbic loops [17,18]. This association remained significant after adjusting for disease duration and disease severity as covariates [19]. Although anxiety and depression are often present throughout (or even before) PD, it has been reported that more severe depressive symptoms (as assessed by the Hamilton Depression Rating Scale) and anxiety (as evaluated by the Hamilton Anxiety Rating Scale) might correlate with a higher risk of FOG in PD [11,14,20].

## 2. Clinical Features of FOG

FOG is known to be one of the most debilitating motor impairments in PD, as it generally hinders mobility, leading to more falls and injuries. FOG is defined as a brief episodic absence or any marked reduction in the forward progression of the feet [21]. Notably, FOG is not exclusive to PD, being present in atypical parkinsonisms and as primary progressive FOG (PPFOG). This latter condition was identified several years ago, and clinical diagnostic criteria were proposed [22]. PPFOG lacks the motor signs and non-motor symptoms that are typically present in PD and other parkinsonisms and does not improve with levodopa therapy, showing mild and dubious responses to amantadine, selegiline, and rasagiline [23,24,25]. Most patients end up wheelchair-bound within 5 years of onset. [26] Neuropathological evidence suggests that PPFOG is not a distinct entity but a syndrome with diverse causes, with possible different evolutions after several years from its clinical onset [27].

FOG mainly occurs in paroxysmal episodes, with a tendency to disappear when patients shift from an automatic control of gait toward a more goal-directed one [28,29]. In other words, gait usually can improve whenever patients consciously focus on walking. Typically, each episode can last from a few seconds up to more than 10 s. Despite being associated with a higher severity of disease, FOG symptoms have not been correlated with the primary features of PD: tremor, bradykinesia, and rigidity [30]. However, the higher severity of FOG is associated with falls, postural instability, and executive dysfunction [31]. Moreover, the gait of PD patients with FOG often shows the following features: high variability in step time, reduced pendular limb movements, reduced stride amplitude, and increased cadence, especially during turns [32]. Episodes are usually triggered by actions such as turning (especially in confined spaces), initiating walking, passing through narrow areas like doorways, or walking while performing another task [33]. As reported, anxiety can contribute to the triggering of FOG [34]. In terms of cognitive and psychological profiles and the severity of FOG, the literature presents controverting data, with a tendency of not finding significant and consistent differences in cognitive function between PD patients with FOG and those without, suggesting that standard neuropsychological tests may not be sensitive enough to detect cognitive discrepancies between these two groups [35,36]. Neuroimaging combined with more targeted neuropsychological tests for striatal function might be able to shed light on these aspects in the future. Nonetheless, significant differences have been reported in neuropsychiatric and clinical symptoms, with a higher prevalence of anxiety, delusion, and impulse control disorder [37].

FOG can manifest various patterns, which can be categorized by the following types of leg movements: (i) FOG characterized by alternating leg trembling at a frequency of 3 to 8 Hz; (ii) FOG where patients take very small shuffling steps; and (iii) an akinetic form of FOG where the legs do not move at all [38]. Another proposed classification addresses the following potential neurobiological causes and associated triggers: (i) motor impairments, such as difficulty turning on the spot; (ii) heightened anxiety, which can increase freezing episodes when the person is in a hurry; and (iii) attentional deficits, such as difficulties with dual-tasking while walking [39].

Several studies performed in gait laboratories have sought to characterize FOG using motion analysis to measure body kinematics, foot switches, force plates, and surface electromyography (EMG) in order to best dissect lower-limb muscle activity [40,41,42,43]. More recently, wearable inertial sensors have been employed to characterize the episodes of FOG [41,42,44,45], providing the possibility of detecting early signs and patterns of gait deterioration before a diagnosis of FOG [46].

## 3. Botulinum Toxin in Parkinson’s Disease

Botulinum neurotoxin (BoNT) has been found to have many applications in treating several motor and non-motor symptoms [47].

BoNT injections can effectively reduce the severity of tremors in PD patients, leading to an improvement in their daily activities and fine motor skills [48]. Common injection sites include the flexor carpi ulnaris, flexor carpi radialis, extensor carpi ulnaris, extensor carpi radialis, biceps, triceps, and the supinator. While upper extremity weakness is a potential side effect, new techniques like the Yale protocol and kinematic tremor analysis can be effective treatments without increasing the risk of hand weakness [49].

BoNT injections are the primary treatment for dystonia in PD. For Pisa syndrome, BoNT-A injections have been shown to significantly improve lateral bending, though results have varied across studies [50]. BoNT-A has also proven to be effective in treating camptocormia [51].

Regarding BoNT-A treatment for pain in PD, most of these studies have reported significant improvements in pain scores and clinical global impressions at 4- and 12-weeks post-injection, particularly for dystonic pain [50].

BoNT has also proven to be an effective treatment for sialorrhea, with both BoNT-A and BoNT-B providing benefits. In fact, up to 80% of patients treated with BoNT-A saw significant improvements [52]. Submandibular gland injections were reported to be more effective than parotid gland injections, although no significant difference was found in UPDRS-drooling scores between the two sites [53]. BoNT-B has also been associated with positive outcomes, leading to improvements in DFSS, global impression scores, Visual Analogue Scale (VAS)-drooling, and Drooling Rating Scale scores [54]. Additionally, BoNT-B has been observed to provoke a quicker onset of action when compared to BoNT-A [55]. Reported adverse effects have been rare and have included dry mouth and saliva thickening, which is more common after submandibular injections [53].

Urodynamic studies have shown that most PD patients will have detrusor muscle overactivity, characterized by low uninhibited detrusor contraction (UDC) at first volume and high UDC at maximum pressures. Though the effects of BoNT-A treatment on UDC at first volume have been mixed, most of these studies indicated a significant increase in maximum cystometric capacity, suggesting improved bladder function [56]. Overall, patients referred to marked improvements in their quality of life, with low rates of adverse effects, such as urinary tract infections, urinary retention, or systemic side effects, being reported during the studies [47].

## 4. Botulinum Toxin for Freezing of Gait

BoNT is widely used in the treatment of PD [49]. The advantages of BoNT, compared to oral medications, include localized action and a low incidence of systemic side effects [57]. These features are important in treating patients with PD, since patients are often older with multiple comorbidities and are frequently on multiple medications [58].

The peculiarity of BoNT relies on its capability to act in a wide range of neural pathways (motor, autonomic, and sensory), thus producing different effects. It is well known that when BoNT-A enters a nerve terminal, it cleaves and prevents the SNARE proteins from assembling the SNARE, thus impeding ACh release from the presynaptic terminal into the neuromuscular junction. By doing so, BoNT reduces muscle contraction by inducing a decreased muscle tone in the legs to improve mobility. Clinically, there are some similarities between FOG and dystonic gait [59,60]. Both conditions can be influenced by behavioral modifications, sensory input, or motor tricks, and tend to occur in specific “automatic” walking patterns, which may involve spastic muscle dysfunction that impairs the initiation of walking. While the neuroanatomical mechanisms underlying FOG remain uncertain, FOG may share a common underlying mechanism with eyelid-opening apraxia (a form of dystonia), which responds to BoNT-A treatment. This insight provided the basis for exploring the use of BoNT-A in FOG. Previous research has indicated that one contributing factor to frozen gait is the lack of synchronized muscle contractions in the legs, including the gastrocnemius [40]. Based on this evidence and the established use of BoNT-A for treating dystonia, it was hypothesized that, in addition to their muscle-relaxing properties, BoNT-A injections could potentially reorganize muscle activity patterns by affecting afferent pathways from the injection site, possibly originating from muscle spindles, and act as a long-term “sensory trick” [61]. In fact, research suggests that, in addition to its local muscle-relaxing effects, BoNT-A may also influence central mechanisms [62]. By reducing abnormal, dysregulated muscle contractions (such as those observed in the gastrocnemius and other lower limb muscles during gait), BoNT could potentially reorganize motor control patterns [63]. Some theories propose that this reorganization might occur via afferent sensory pathways, which could help “reset” abnormal motor feedback, reducing FOG episodes. There is also evidence from animal models indicating that BoNT might act on central motor circuits, including the striatal system, which is critical in the regulation of movement in PD [64].

BoNT-A could also indirectly play a role in alleviating FOG by modulating neuroinflammation. Aside from BoNT-A’s well-established and broad use in neuropathic pain modulation [65,66,67,68,69,70], limited evidence suggests that BoNT-A could be involved in the regulation of the inflammatory process, apparently not limited to the nervous system [71,72,73,74]. Using data from more than 200 patients from the COPPADIS-2015 study, authors suggested that the serum’s ultra-sensitive C-reactive protein level could be related to FOG in PD patients, showing significantly higher levels in patients with FOG.

In a small exploratory study, Hatcher-Martin and colleagues analyzed cerebrospinal fluid biomarkers in PD patients with FOG and reported lower levels of fractalkine, an anti-inflammatory protein that, under some circumstances, contributes to suppressing microglial activation and maintaining the microglia surveillance phenotype [75]. This evidence might indicate a possible role of neuroinflammation in the pathophysiology of FOG in PD, although further studies are required to shed light on these novel central mechanisms. By doing so, BoNT could rely on a wider body of evidence to broaden its field of application and possibly contribute to the reduction in FOG in PD.

To date, studies have investigated botulinum toxin for the treatment of FOG in parkinsonism. All of the seven included studies in our review differed in terms of methodology, design, dosage, and outcome (Table 1). Concerning the site of injection, four studies reported that the soleus gastrocnemius complex, mono or bilaterally, were the treated muscles. In these reported cases, the authors utilized different dosages, including 175 for one leg, 100–300 IU for mono or bilateral treatment, and 150–200 bilaterally, as well as 5000 IU of BoNT-B for the monolateral soleus/gastrocnemious. Two studies used a peculiar scheme of treatment, injecting the tensor fasciae latae and the psoas major muscles. In one of these studies, the authors utilized 50 IU for the treatment of tensor fasciae latae, while in the second study, the dosage was not specified. Regarding the recruited patients, six out of seven studies diagnosed PD patients with FOG. In Giladi et al. 2001 [10], a single case of vascular parkinsonism and a case of pure FOG were included.

Generally, treatment was performed under electromyography guidance. However, ultrasonography guidance may enhance the visualization of BoNT injections into the muscles [76,77] and consequently better define the muscle groups (Figure 1).

Several systems of clinical evaluation have beenutilized: the UPDRS, the Hoehn and Yahr Scale, the Timed Up and Go Test, the Freezing of Gait Questionnaire, the Clinical Global Impression Scale, the VAS, the modified-Webster Step-ST, the Subjective Clinical Global Impression of Change scale, and the 39-Item PD Questionnaire. A video recording was employed in one study, and in another, a BOLD-fMRI investigation was carried out.

Finally, as for the study design, 3 out of 6 of these studies were open-label, while the remaining 3 were double-blind and placebo-controlled. Only the controlled studies reported no benefit from the treatment.

**Table 1 toxins-16-00474-t001:** Included studies on the treatment with botulinum toxin of freezing of gait.

	Authors	Date	Study	Disease	Disease Duration (Mean ys)	Number of Patients	Mean Age	Clinical Evaluation	Type of FOG	Procedure	Type of Toxin	Total Dosage (I.U.)	Sites	Effect (Y/N)	Duration of Effect	Adverse Effect
1	Giladi N et al. [10]	2001	OL	8 PD, 1 VP, I pure FOG	13.9	10	70.4	SCGIG scale	On and off	EMG guidance	BoNT-A	100–300	mono/bilateral ELA and SG, TP	Y	2–12 weeks	n/s
2	Fernandez HH et al. [78]	2004	DB, PC	PD	10	9 (5 ctr)	74	UPDRS II-III, VAS, CGIS, mWebster Step-ST, videotape	n/s	clinical identification	BoNT-B	5000	monolateral SG	N		dry mouth (2), increased festination (1)
3	Wieler M et al. [79]	2005	R, DB, PC, CO	PD	12	12	67	FOG-Q, H&Y, UPDRS, TUG, PDQ-39	n/s	EMG guidance	BoNT-A	200–300	bilateral SG	N		n/s
4	Gurevich T et al. [61]	2007	P, DB, PC, CO	PD	10.4	11	69.4	FOG-Q, UPDRS, ADL, SCGIC	Off	EMG guidance	BoNT-A	300 (150 per leg)	bilateral SG	N		leg weakness (3/6), increased falls (2/6)
5	Vastik et al. [80]	2016	OL	PD	5–15	11	71.2	BOLD fMRI, FOG-Q, TUG, CGSI	n/s	EMG guidance	BoNT-A	100 (50 per leg)	bilateral TFL	Y	n/s	n/s
6	Neshige R [81]	2022	OL	PD	n/a	5	74.4	UPDRS, FOG-Q, TUG	n/s	not specified	BoNT-A	n/s	bilateral PM	Y	From 1 to 6 months	n/s

ADL: activities of daily living; BoNT-A: botulinum toxin type A; BoNT-B: botulinum toxin type B; CGIS: Clinical Global Impression Scale; CO: cross-over; DB: double-blind; ELA: extensor longus hallucis; EMG: electromyografy; FOG: freezing of gait; FOG-Q: freezing of gait questionnaire; n/s: not specified; OL: Open Label, PD: Parkinson’s disease; PC: placebo-controlled, PM: psoas major; SCGIG: Subjective Clinical Global Impression of Change scale; SG: soleus gastrocnemius; TFL: tensor fasciae latae; TUG: Timed Up and Go Test; UPDRS: Unified Parkinson’s disease rating Scale; VAS: Visual Analogue Scale; VP: vascular parkinsonism; Y/N: Yes/Not.

## 5. Conclusions 

Our review findings on past studies that specifically investigated the effects of botulinum toxin on FOG in Parkinson’s disease found that the results between open-label and double-blind studies were conflicting in terms of outcome. However, given that all the investigating studies in both groups included sample sizes that were too small, the results cannot be deemed reliable for clinical practice. Future studies will need to have large sample sizes to obtain the necessary amount of data to reach reliable conclusions.

## 6. Methods

A comprehensive literature search was performed on PubMed combining the following search terms: “botulinum toxin”, “botulinum”, “freezing of gait”, and “FOG”. The search was limited to articles published between January 1, 1994 and September 1, 2024. The search was restricted to the English language; viewpoints were included as well as original articles. The initial search yielded 22 articles; however, only 6 of these specifically investigated the effects of botulinum toxin on freezing of gait in Parkinson’s disease and met the inclusion criteria (Figure 2).

## Figures and Tables

**Figure 1 toxins-16-00474-f001:**
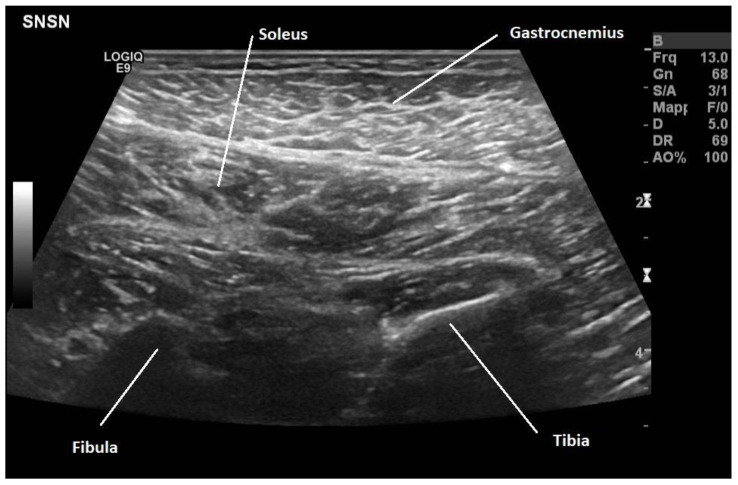
Ultrasound image of the gastrocnemius and soleus muscles.

**Figure 2 toxins-16-00474-f002:**
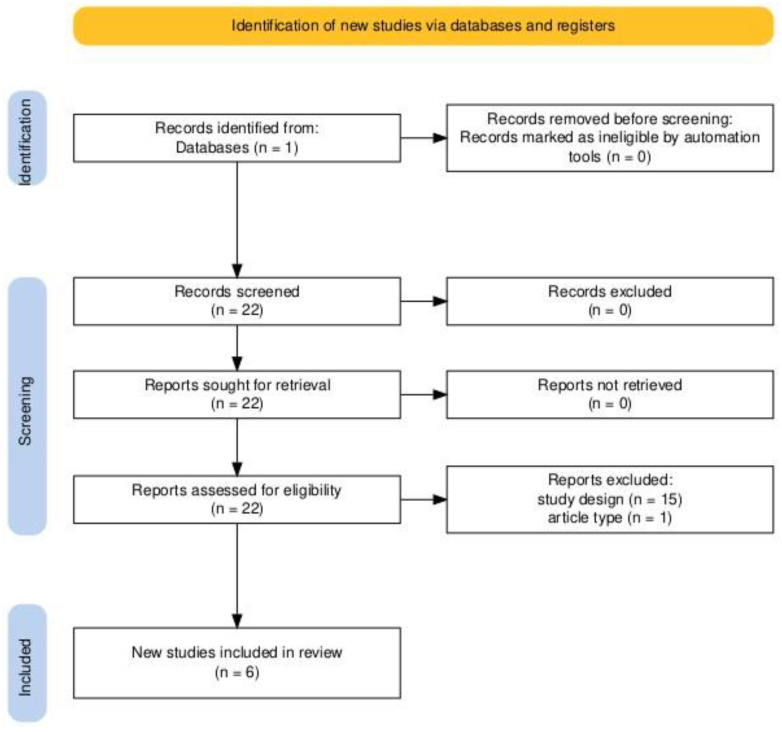
PRISMA flowchart.

## Data Availability

No new data were created or analyzed in this study. Data sharing is not applicable to this article.

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
