# Peer review of "Botulinum Toxin Effects on Freezing of Gait in Parkinson’s Disease: A Systematic Review"

_toxins, 2024, doi:10.3390/toxins16110474_

Round 1

Reviewer 1 Report

Comments and Suggestions for Authors

1.     The review would benefit from a more detailed discussion of the mechanisms through which BoNT may or may not affect FOG in PD patients. Additionally, an exploration of why certain studies found no benefit from BoNT would help to contextualize the conflicting results.

2.     I think it would be extremely interesting to discuss if the role of BoNT in inhibiting neurogenic inflammation, mainly by impeding nerve-derived release of CGRP and SP, could be useful in patients with parkison disease and Freezing of Gait?

Please see this paper: Popescu MN, Beiu C, Iliescu MG, Mihai MM, Popa LG, Stănescu AMA, Berteanu M. Botulinum Toxin Use for Modulating Neuroimmune Cutaneous Activity in Psoriasis. Medicina. 2022; 58(6):813. https://doi.org/10.3390/medicina58060813 

From my point of view, though FOG is primarily viewed as a motor symptom, there are several points worth considering regarding neurogenic inflammation and its modulation by BoNT:

·      Modulation of Sensory and Proprioceptive Input: The sensory system and proprioceptive feedback are critical for gait control. If neurogenic inflammation contributes to altered sensory input in PD, inhibiting CGRP and SP release could theoretically help in normalizing sensory signals and improving gait patterns.

·      Non-Motor Pathways: Some studies have shown that non-motor factors, including anxiety and cognitive impairments, can trigger or worsen FOG. By modulating neurogenic inflammation, BoNT may influence these pathways, potentially reducing the occurrence of FOG in some patients.

Author Response

Reviewer 1:

Comment 1:  The review would benefit from a more detailed discussion of the mechanisms through which BoNT may or may not affect FOG in PD patients. Additionally, an exploration of why certain studies found no benefit from BoNT would help to contextualize the conflicting results.

 Response 1: We thank the reviewer for this observation. We specified in the revised draft that:

“The peculiarity of BoNT relies on its capability to act in a wide range of neural pathways (motor, autonomic and sensory) thus producing different effects. It is well known that when BoNT-A enters a nerve terminal, it cleaves and prevents the SNARE proteins from assembling the SNARE, thus impeding ACh release from the presynaptic terminal into the neuromuscular junction.

By doing so, BoNT reduces muscle contraction by inducing a decreased muscle tone in the legs to improve mobility. Clinically, there are some similarities between FOG and dystonic gait. [51,52] Both conditions can be influenced by behavioral modifications, sensory input, or motor tricks, and tend to occur in specific "automatic" walking patterns, which may involve spastic muscle dysfunction that impairs the initiation of walking. While the neuroanatomical mechanisms underlying FOG remain uncertain, FOG may share a common underlying mechanism with eyelid opening apraxia (a form of dystonia), which responds to BoNT-A treatment. This insight provided the basis for exploring the use of BoNT-A in FOG, Previous research has indicated that one contributing factor to frozen gait is the lack of synchronized muscle contractions in the legs, including the gastrocnemius. Based on this evidence and the established use of BoNT-A for treating dystonia, it was hypothesized that, in addition to its muscle-relaxing properties, BoNT-A injections could potentially reorganize muscle activity patterns by affecting afferent pathways from the injection site, possibly originating from muscle spindles, and act as a long-term "sensory trick.". the muscle contraction by inducing a decreased muscle tone in the legs, in order to improve mobility.  In fact, research suggests that in addition to its local muscle-relaxing effects, BoNT-A may also influence central mechanisms. By reducing abnormal, dysregulated muscle contractions (such as those observed in the gastrocnemius and other lower limb muscles during gait), BoNT could potentially reorganize motor control patterns. Some theories propose that this reorganization might occur via afferent sensory pathways, which could help "reset" abnormal motor feedback, reducing FOG episodes. There is also evidence from animal models indicating that BoNT might act on central motor circuits, including the striatal system, which is critical in the regulation of movement in PD.”

Comment 2: I think it would be extremely interesting to discuss if the role of BoNT in inhibiting neurogenic inflammation, mainly by impeding nerve-derived release of CGRP and SP, could be useful in patients with parkison disease and Freezing of Gait?

Please see this paper: Popescu MN, Beiu C, Iliescu MG, Mihai MM, Popa LG, Stănescu AMA, Berteanu M. Botulinum Toxin Use for Modulating Neuroimmune Cutaneous Activity in Psoriasis. Medicina. 2022; 58(6):813. https://doi.org/10.3390/medicina58060813 

From my point of view, though FOG is primarily viewed as a motor symptom, there are several points worth considering regarding neurogenic inflammation and its modulation by BoNT:

  • Modulation of Sensory and Proprioceptive Input: The sensory system and proprioceptive feedback are critical for gait control. If neurogenic inflammation contributes to altered sensory input in PD, inhibiting CGRP and SP release could theoretically help in normalizing sensory signals and improving gait patterns.
  • Non-Motor Pathways: Some studies have shown that non-motor factors, including anxiety and cognitive impairments, can trigger or worsen FOG. By modulating neurogenic inflammation, BoNT may influence these pathways, potentially reducing the occurrence of FOG in some patients.

Response 2: To clarify this important aspect, we added our interpretation, based on literature, regarding the role of BoNT in inflammation.

BoNT-A could also indirectly play a role in alleviating FOG by modulating neuroinflammation. Aside from   BoNT-A’s well-established and broad use in neuropathic pain modulation, limited evidence suggests that BoNT-A could be involved in the regulation of inflammatory process, apparently not limited to the nervous system. Using data from more than 200 patients from the COPPADIS-2015 study, authors suggested that serum ultra-sensitive C reactive protein level could be related to FOG in PD patients, showing significantly higher levels in patients with FOG.

In a small exploratory study, Hatcher-Martin and colleagues analyzed cerebrospinal fluid biomarkers in PD patients with FOG and reported lower levels of fractalkine, an anti-inflammatory protein that under some circumstances contributes to suppressing microglial activation and maintaining the microglia surveillance phenotype. This evidence might indicate a possible role of neuroinflammation in the pathophysiology of FOG in PD, although further studies are required to shed light on these novel central mechanisms. By doing so, BoNT could rely on a wider body of evidence to broaden its field of application and possibly contribute to the reduction of FOG in PD.”

Reviewer 2 Report

Comments and Suggestions for Authors

Most of the material in the paper such as epidemiology of FOG, other treatments etc is irrelevant.

This should be a short letter to the editor at best that the evidence for BoNTA in FOG is lacking. 

Nothing new in this paper.

Comments on the Quality of English Language

Needs editing for English

Author Response

Reviewer 2:

Comment 1: Most of the material in the paper such as epidemiology of FOG, other treatments etc is irrelevant. This should be a short letter to the editor at best that the evidence for BoNTA in FOG is lacking.  Nothing new in this paper.

Response 1: Thanks for the suggestions. I hope the revised version may be more interesting in pointing out the fact that there is no clear evidence of benefits of BoNT on freezing of gait and that the specialists needs further well-done studies for its demonstration.  

Reviewer 3 Report

Comments and Suggestions for Authors

This review paper on BoNT treatment in PD patients with FOG is very well written. But I have notice that authors did not described Method section. They did not wrote which scientific bases they have searched, how many studies they have found, which of them were exclude and why. This procedure also need flow diagram. 

In this part "In these reported cases, the authors utilized different dosages -175 for one leg, to 100- 157 300 Units for mono or bilateral treatment, 150/200 bilaterally as well as 5000 U. BoNT-B 158 for monolateral soleus/gastrocnemious. Two studies used a peculiar scheme of treatment, 159 injecting the tensor fasciae latae and the psoas major muscles. In one of these studies, the 160 authors utilized 50 U.I. for the treatment of tensor fasciae latae, while in the second study 161 the dosage was not specified" authors used different abbreviation for IU (U.I.; U.; units). Please be uniform with this and use just one abbreviation. 

Author Response

Reviewer 3

Comment 1:  This review paper on BoNT treatment in PD patients with FOG is very well written. But I have notice that authors did not described Method section. They did not wrote which scientific bases they have searched, how many studies they have found, which of them were exclude and why. This procedure also need flow diagram.

Response 1: We revised the manuscript, according to the suggestions. Particularly, I am pleased to inform you we specified in the Methods using PRISMA guideline and added the PRISMA flowchart (fig.1).

Comment 2: In this part "In these reported cases, the authors utilized different dosages -175 for one leg, to 100- 157 300 Units for mono or bilateral treatment, 150/200 bilaterally as well as 5000 U. BoNT-B 158 for monolateral soleus/gastrocnemious. Two studies used a peculiar scheme of treatment, 159 injecting the tensor fasciae latae and the psoas major muscles. In one of these studies, the 160 authors utilized 50 U.I. for the treatment of tensor fasciae latae, while in the second study 161 the dosage was not specified" authors used different abbreviation for IU (U.I.; U.; units). Please be uniform with this and use just one abbreviation. 

Response 2: We thank the referee for the suggestion. Accordingly, in the revised draft the dosages have been corrected.

Reviewer 4 Report

Comments and Suggestions for Authors

It is somewhat  interesting ,  however Botulinum toxin  is usually block muscular  strength,therefore to aim at some parts of   freezing of gait in Parkinson’s disease, but all cases. This review is essential and deserve to publish it.

The authors firstly discuss the concept of  patients with primary progressive freezing of gait, as the prodrome of PD, or others, e.g.. PSP.

Some neuropsychological factors may affect these outcome expression, not the true effects of B-toxin.

PLease add some papers into the Reference.

Zhang L L , Zhao Y J , Zhang L,  ,et al. Experience of diagnosis and managements for patients with primary progressive freezing of gait[J]. Journal of Neurorestoratology, 2023, 11(1):100039-100039. 

Sun H, Gan C, Wang L, Ji M, Cao X, Yuan Y, Zhang H, Shan A, Gao M, Zhang K. Cortical Disinhibition Drives Freezing of Gait in Parkinson's Disease and an Exploratory Repetitive Transcranial Magnetic Stimulation Study. Mov Disord. 2023 Nov;38(11):2072-2083.  

Taximaimaiti R, Wang XP. Comparing the Clinical and Neuropsychological Characteristics of Parkinson's Disease With and Without Freezing of Gait. Front Neurosci. 2021 Apr 27;15:660340. 

Comments on the Quality of English Language

OK

Author Response

Reviewer 4:

Comment 1:  The authors firstly discuss the concept of patients with primary progressive freezing of gait, as the prodrome of PD, or others, e.g.. PSP.

Response 1: We thanks the referee for the comment. We revised the manuscript, according to the suggestions and added some sentences regarding this issue.

Comment 2:  Some neuropsychological factors may affect these outcome expression, not the true effects of B-toxin. Please add some papers into the Reference.

Zhang L L , Zhao Y J , Zhang L,  ,et al. Experience of diagnosis and managements for patients with primary progressive freezing of gait[J]. Journal of Neurorestoratology, 2023, 11(1):100039-100039.

Sun H, Gan C, Wang L, Ji M, Cao X, Yuan Y, Zhang H, Shan A, Gao M, Zhang K. Cortical Disinhibition Drives Freezing of Gait in Parkinson's Disease and an Exploratory Repetitive Transcranial Magnetic Stimulation Study. Mov Disord. 2023 Nov;38(11):2072-2083. 

Taximaimaiti R, Wang XP. Comparing the Clinical and Neuropsychological Characteristics of Parkinson's Disease With and Without Freezing of Gait. Front Neurosci. 2021 Apr 27;15:660340.

 Response 2: The reference paragraph has been completed with the suggested papers.

Round 2

Reviewer 3 Report

Comments and Suggestions for Authors

I have no further comments